# Effects of *Trichoderma harzianum* Strain T22 on the Arthropod Community Associated with Tomato Plants and on the Crop Performance in an Experimental Field

**DOI:** 10.3390/insects13050418

**Published:** 2022-04-28

**Authors:** Vittoria Caccavo, Pierluigi Forlano, Stefania Mirela Mang, Paolo Fanti, Maria Nuzzaci, Donatella Battaglia, Vincenzo Trotta

**Affiliations:** 1School of Agricultural, Forestry, Food and Environmental Sciences (SAFE), University of Basilicata, Viale dell’Ateneo Lucano 10, 85100 Potenza, Italy; vittoria.caccavo@unibas.it (V.C.); pierluigiforlano90@gmail.com (P.F.); stefania.mang@unibas.it (S.M.M.); maria.nuzzaci@unibas.it (M.N.); donatella.battaglia@unibas.it (D.B.); 2Department of Science, University of Basilicata, Viale dell’Ateneo Lucano 10, 85100 Potenza, Italy; paolo.fanti@unibas.it

**Keywords:** *Trichoderma*, field experiment, QBSar, plant pathogens, tomato aphids, *Chaetocnema tibialis*, *Tetranychus urticae*, leaf miner, integrated pest management

## Abstract

**Simple Summary:**

There is currently a global research interest in reducing off-farm input of chemical pesticides and fertilizers by using green alternative practices. Fungi belonging to the genus *Trichoderma* colonize plant roots and activate systemic plant defenses against the attack of pests and pathogens. *Trichoderma* spp. have positive impacts on the environment and guarantee food security, which, in turn, offer important economic benefits in agriculture. The purpose of the present study was to investigate the effects of the inoculation of a commercial *Trichoderma* strain on the arthropod community, downy mildew, and the agronomic performance of tomato plants in an experimental field. Our results showed that inoculation with *Trichoderma* positively influenced tomato fruit yields and could reduce the abundance of specific pests under field conditions.

**Abstract:**

Fungi belonging to the genus *Trichoderma* have received much attention in recent years due to their beneficial effects on crop health and their use as pest control agents. *Trichoderma* activates direct plant defenses against phytophagous arthropods and reinforces indirect plant defense through the attraction of predators. Although the plant defenses against insect herbivores were demonstrated in laboratory experiments, little attention has been paid to the use of *Trichoderma* spp. in open field conditions. In the present study, we investigated the effects of the inoculation of the commercial *Trichoderma harzianum* strain T22 on the arthropod community associated with tomato plants and on the crop performance in an experimental field located in South Italy. Our results showed that inoculation with *T. harzianum* could alter the arthropod community and reduce the abundance of specific pests under field conditions with respect to the sampling period. The present study also confirmed the beneficial effect of *T. harzianum* against plant pathogens and on tomato fruit. The complex tomato–arthropod–microorganism interactions that occurred in the field are discussed to enrich our current information on the possibilities of using *Trichoderma* as a green alternative agent in agriculture.

## 1. Introduction

Tomato (*Solanum lycopersicum* L.) is one of the most important vegetable crops cultivated in the world, second only to potato, and one of the basic foods in the Mediterranean diet [1,2]. Fresh and processed tomatoes are widely consumed in the Mediterranean area, and Italy is one of the main producers and suppliers of processed tomatoes in the world [3,4,5]. Agrochemicals, such as synthetic fertilizers, pesticides, and herbicides, are usually used in tomato production to maximize yield and product quality and to achieve low production costs [6,7]. However, their overuse can cause environmental pollution and human health problems. There is currently a global interest in reducing off-farm input of chemical pesticides and fertilizers by using green alternative practices. Among all alternatives, numerous biological products based on beneficial plant microbes, such as bacteria (*Bacillus, Pseudomonas*) [8,9] or fungi (*Trichoderma*, *Beauveria,* mycorrhizae) [10,11,12,13], are receiving a lot of attention in agricultural farming systems due to their valuable properties. These products are used for pest control and for their potential to increase crop health and fitness; they also do not have any negative impacts on the environment and guarantee food security.

Fungi belonging to the genus *Trichoderma* have received much attention in recent years due to their beneficial effects on host plants [14,15,16,17,18,19]. These fungi are distributed throughout the world and are capable of colonizing plant roots and establishing chemical communication with the host plant [20]. Moreover, *Trichoderma* spp. have many possible uses and were investigated for their direct effects on the host plants, such as the increase in nutrient uptake, efficiency of nitrogen use, and seed germination rate, which offer important economic benefits in agriculture. These fungi also promote plant growth and resistance against biotic and abiotic stresses [14,21,22]. It was found that the plant systemic acquired resistance (SAR) and/or the induced systemic resistance (ISR) against biotic and abiotic stress agents could be activated by some *Trichoderma* strains [14,21,23].

Following the *Trichoderma* roots colonization, the plant reacts when it is attacked by a potential root endophytic pathogen, thus activating local and systemic defense mechanisms. Therefore, the plant limits the fungus penetration inside the root, restoring its integrity and antimicrobial activity to the pre-infection levels [20,21,23]. Once this equilibrium is reached, the plant receives protection and more available nutrients, while the fungus receives organic compounds. In this way, *Trichoderma* activates systemic plant defenses against the attack of pests and/or pathogens.

The feeding activity of phytophagous insects also elicits the release of attractive compounds [24]. Volatile organic compound (VOC) blends released in response to a pest attack have direct and indirect defensive effects on insect performance [25]. Plant responses to herbivores, induced by the different modes in which these organisms attack the plant, were shown to be affected by *Trichoderma* colonization [26,27,28,29]. Several studies were focused on the role of *Trichoderma* spp. (and their metabolites) on multitrophic interactions or on plant growth and defence responses [16,18,30,31].

Although direct and indirect plant defenses against insect herbivores were demonstrated in different plant species in greenhouses or pot experiments, little attention has been paid to the use of *Trichoderma* spp. in open field conditions [16]. The interaction between the plant and *Trichoderma* spp. directly confers some degree of protection against nematodes [32] and insects, such as aphids [33,34], thrips [26], and caterpillars [27,33]. For example, the survival of the aphid *Macrosiphum euphorbiae* (Thomas) in tomato plants could be significantly reduced using the P1 strain of *T. atroviride* as a consequence of the upregulation of genes involved in the oxidative burst reaction [33]. Similar results were obtained when the *T. harzianum* strain T22 was used in the same context [34]. The tomato defense responses against the green stink bug *Nezaria viridula* L. were enhanced by the *T.*
*harzianum* strain T22 through an early increase in transcript levels of JA marker genes [35]. In onions, the performance of *Thrips tabaci* L. was reduced after colonization by *Trichoderma* spp. [26]. Among the chewing insects, the *T. atroviride* P1 strain was associated with reduced survival and development of *Spodoptera littoralis* (Boisduval) larvae and with an enhanced expression of genes encoding for protective enzymes in tomato plants [33]. Maize inoculation with *T. atroviride* increased plant growth, altered the feeding pattern of *Spodoptera frugiperda* (JE Smith) larvae, and was correlated with an increased emission of volatile terpenes and accumulation of JA [27]. In an in vitro assay, the secondary metabolite 6-pentyl-α-pyrone produced by the bioactivity of *T. asperellum* caused a high mortality rate in the two-spotted spider mite *Tetranychus urticae* Koch [36].

*Trichoderma* spp. also affect above-ground plant–insect interactions, reinforcing indirect plant defense barriers against phytophages through the production and release of VOCs that are involved in the attraction of predators and parasitoids [16,23]. For example, *T. longibrachiatum* MK1 influences the quantity and quality of VOCs released by the tomato plant (such as methyl salicylate), improving the attractiveness and performance of the aphid parasitoid *Aphidius ervi* Haliday and the predator *Macrolophus pygmaeus* (Rambur) [37]. In tomato, colonization by *T. atroviride* P1 significantly increases the attraction of *A. ervi* [33]. In a multitrophic interaction system, *Trichoderma atroviride* IMI 206040 associated with maize roots was shown to increase the parasitism rate of *Campoletis sonorensis* (Carlson) on *S. frugiperda* [38]. As suggested by Battaglia et al. [37], the improved attractiveness and performance of insect predators on *Trichoderma*-colonized plants could be considered a result of the “increased fitness flow” modulated through the interaction between the primary and secondary metabolism of plants [39].

However, below- and above-ground plant–insect–microorganism interactions are very complex and may be very different under field conditions. Contreras-Cornejo et al. [40] showed that, in a maize field, the community of native foliage arthropods could be altered after plant inoculation with *T. harzianum* strain 38. The authors found that the number of arthropods per plant did not differ between the inoculated and control plants. Nevertheless, *T. harzianum* inoculation decreased the number of piercing-sucking insects but increased the abundance of chewing herbivores and predators. The presence of *Trichoderma* was also shown to influence JA-mediated VOCs production in a vineyard, attracting parasitoid wasps of the Mymaridae family [41].

*Trichoderma harzianum* can also modulate soil arthropod biodiversity. For example, a higher abundance of collembolans was found under optimal conditions but not under suboptimal or adverse ones [42].

In many cases, the outcomes of plant–*Trichoderma* interactions are species-specific and even strain-specific [20,43,44,45,46]. *Trichoderma*–plant interactions with pests and their natural enemies are influenced by environmental conditions when experiments are performed in the field. Recently, it was demonstrated that the tomato defense response against insect pests induced by diverse *Trichoderma* species is influenced by the temperature [47]. Therefore, the use of *Trichoderma* as a biocontrol agent in agriculture depends not only on the targeted use (pests and pathogens) but also on local climatic conditions, soil properties, the availability of water and nutrients, and crop species.

The aim of the present study was to investigate the effects of the inoculation of a commercial *Trichoderma harzianum* strain T22 on the arthropod community associated with tomato plants in an experimental field located in South Italy. The beneficial effects of *T. harzianum* on the soil arthropod biodiversity, as well as on the agronomic performance of tomato plants (improved yield) and the behavior toward downy mildew (one of the main phytopathogens of tomato in South Italy) were also investigated. The complex tomato–arthropod–microorganism interactions that occurred in the field are also discussed to enrich our current information on the possibilities of using *Trichoderma* as a green alternative agent in agriculture.

## 2. Materials and Methods

### 2.1. Crop Cultivation

The present study was performed in an experimental tomato field located in Pignola (40°34′06.2″ N, 15°45′35.4″ E; 780 m above sea level), Potenza, Italy, during the 2021 tomato growing season. According to the FAO world reference base for soil resources, the soil was a dystric cambisol (Bd68-2bc), with the following characteristics: particles smaller than 2 mm in size, 935 g/kg; particles larger than 2 mm, 65 g/kg; apparent density, 1.294 kg/dm^3^; texture composition of sand, 481 g/kg; clay, 149 g/kg; silt, 370 g/kg at depth of 0–30 cm. The contents of total carbonate and total organic matter were 16 g/kg and 32.8 g/kg, respectively. The composition of the soil was as follows: total N, 2 g/kg; P, 29 mg/kg; Ca 11.1 meq/100 g; Mg, 4.6 meq/100 g; Na, 1.8 meq/100 g; soil pH (H_2_O), 6.2.

The soil was left fallow the year before the experiment and then plowed to a depth of 30 cm, rotavated, and leveled before planting the crop. Tomato seedlings (*Solanum lycopersicum* L.) of the commercial cultivar San Marzano Kero were used in this experiment. Tomato plants placed in alveolate containers were purchased from a nursery and were transplanted into the field on 1 June 2021. Fertigation was carried out with ammonium nitrate (YaraTera© AMNITRA^TM^, N 34.2%) applied three times in the recovery and blossoming stage (20 kg ha^−1^ at an interval of one week from each other), calcium nitrate (YaraTera© CACINIT^TM^, CaO 26.5%, N 15.5%) applied twice during the fruit-bearing stage (30 kg ha^−1^ at an interval of one week from each other) and potassium nitrate (YaraTera© KRISTA K^TM^ plus, N 13.7%, K_2_O 46.3%) applied once (20 kg ha^−1^) at the fruit-enlargement stage. The use of fertilizers was developed during the present experiment and was appropriate to the needs of the plants, considering the soil characteristics. The tomato plants were not treated with pesticides during the entire field trial.

### 2.2. Meteorological Data

The Agrometeorological Service of the Agenzia Lucana per lo Sviluppo e l’Innovazione in Agricoltura (ALSIA) of the Basilicata Region provided the meteorological data for the area in which the experimental farm is located. The temperature and rainfall data recorded during the experiment are shown in Figure 1. During the period of interest, the average temperature remained less than or equal to 20 °C. The temperatures reached a maximum of 30 °C in August. Precipitations recorded in July, August, and September were low.

### 2.3. Experimental Design

Two treatments were compared: non-inoculated tomato plants (control) and inoculated tomato plants with *Trichoderma harzianum* strain Rifai KRL-AG2 biocontrol agent (T-22) (KOPPERT B.V., Berkel en Rodenrijs, The Netherlands). Forty-day-old tomato plants were inoculated with *T. harzianum* following the manufacturer’s instructions (1 × 10^9^ cfu/g of viable *T. harzianum* T-22 spores) one week before transplantation. The alveolate containers with 320 seedlings were then watered with 3 g of commercial *T. harzianum* dissolved in 3 liters of water. Treatment was repeated after three days and then the seedlings were transplanted. The control plants were not inoculated.

The experiment was carried out on a strip of soil about 58 m long and 9 m wide, divided into 6 plots of 54 m^2^ (6 m × 9 m) separated from each other by a strip 3.6 m wide left without plants. Thus, 3 plots treated with *T. harzianum* T22 and 3 control plots were obtained, alternating along the length of the field. A 10 m strip around the field was plowed and then left uncultivated throughout the entire duration of the experiment. The plants in each plot were manually transplanted on 1 June 2021 in 5 rows 1.8 m apart from each other. Each row was 6 m long with a plant spacing of 33 cm, with a total of 20 plants/row (100 plants/plot).

Six root samples of control or *T.*-*harzianum*-T22-inoculated plants (two per plot) were taken at the end of the experiment to determine the presence of the fungus. The tomato roots were dissected, mounted on a slide, and then observed with a stereomicroscope. The presence of the fungus was confirmed by the observation of hyphae in the secondary roots.

### 2.4. Insect Sampling

During the first month after transplantation, the tomato seedlings are very small and in a critical phase for their vegetative growth. However, seedlings are susceptible to attack by insect pests, such as aphids and beetles. For this reason, the presence/absence of phytophages was recorded via direct observation of the seedlings, without damaging them, on two dates (22 June and 1 July). Ten plants were randomly sampled within each plot, with a total of 30 plants per treatment/date. The presence/absence of insect pests on the plant was recorded; some of the observed specimens were gently removed, placed in an Eppendorf tube filled with ethanol, and transported to the laboratory for identification. On 10 June, yellow sticky traps were randomly placed between two adjacent seedlings in a row (5 traps/plot). After 20 days (1 July), the traps were collected and wrapped in a transparent PVC plastic film. Captured tomato pests were subsequently counted and identified using a stereomicroscope in the laboratory.

After the first month, as the plants were larger, leaf samples were collected for observation in the laboratory under a stereoscopic microscope. This allowed for more accurate sampling of small arthropods, thus obtaining quantitative data on their abundance. Six different samplings were carried out from 12 July to 21 September (more specifically, 12 and 22 July, 3 and 26 August, 8 and 21 September). Within each plot, ten plants were randomly sampled at 9.00 am for a total of 30 plants/date. For each plant, a composite sample was used, consisting of three leaves taken from the apical, middle, and basal parts of the tomato plant. The leaves were detached and placed in plastic cylinders (150 mL). The cylinders were then maintained in darkness at 5 °C and transported to the laboratory for the identification of the arthropods. The arthropods were transferred to 50 mL sterile Falcon tubes, which were filled with ~30 mL of 70% ethanol in water and refrigerated at 4 °C until identification. Individuals were then observed under a stereomicroscope. Arthropods from each sample were classified at the order, family, and, when possible, at the genus and species levels. Furthermore, the presence of damage caused by leaf miners on the leaves was noted and analyzed.

### 2.5. Soil Sampling and Microarthropod Extraction

On 21 September, within each experimental plot, three soil samples were randomly collected between two tomato plants with and without *T. harzianum* T22. A sample was composed of three soil clods (10 × 10 × 15 cm depth), which were taken using a hand auger in three different rows. After collection, soil samples were placed in a plastic bag, kept in darkness at 5 °C, and transported to the laboratory for arthropods extraction. Microarthropod extraction was carried out by gently placing soil clods on mesh-covered funnels (mesh 2 mm, 20 cm in diameter). A plastic jar containing 50 mL of hydroalcoholic solution (70%) was placed at the bottom of the funnel to store the extracted arthropods. Incandescent lamps (40 watts) were placed 20 cm above the soil clods. After 14 days, the extracted specimens were observed under a stereomicroscope, the biological morphs were determined, and the ecological–morphological index (EMI) was assigned. Finally, the QBSar (soil biological quality—arthropod) index was computed as the sum of the EMI values [48,49].

### 2.6. Evaluation of Downy Mildew on Tomato Plants

We focused only on the presence of downy mildew since it is one of the diseases that can cause major damage to tomato in the considered area. All plants were screened for the presence/absence of downy mildew at the beginning of August. For this purpose, each tomato plant was visually inspected and the presence of the abovementioned disease was recorded if at least its initial symptoms were present, such as small pale yellow spots with indefinite borders on the upper leaf surface. At an advanced stage of downy mildew development, other parts of the plant, such as stems, flowers, and fruit, are also attacked, and thus the damage could be very high. The evolution of the downy mildew disease during the entire period of tomato cultivation was also recorded.

### 2.7. Agronomic Performance Estimation

Fruit sampling was performed during the experiment. Within each experimental plot, ten plants were marked and followed throughout their development. Ripe tomato fruit were manually collected from the same plants on three different dates (30 August, 8 and 21 September). All red tomatoes on a plant were harvested, counted, and divided into marketable and non-marketable fruit. Marketable tomatoes were counted, weighed, and measured (maximum length and width), while unmarketable fruit were divided into two groups, rotten and insect-damaged fruit, and finally counted.

### 2.8. Statistical Analysis

Row data used in the analyses of the presence/absence of phytophages on tomato seedlings were the number of plants/experimental plot colonized by insects. Since these data have a discrete probability distribution, a binomial generalized linear model (GLM) with a logit link function was considered the best model for these analyses, thus avoiding transforming the data. For each of the insect species identified, the *p*-values for differences between treatments and sampling dates, as well as their interactions, were obtained through analyses of deviance (type III chi-squared tests). The following model was applied:*Y* = *μ* + *Treatment* + *Date* + *Treatment × Date* + *ε*
where *Y* is the binomial trait studied (number of plants with the presence/absence of a phytophagous), *Treatment* (two levels: *T. harzianum* and control), and *Date* (two levels: 22 June and 1 July) are the fixed effects.

Insect abundances on yellow sticky traps were analyzed via nested analysis of variance (ANOVA) since the homoscedasticity and normality assumptions for ANOVA were checked and met for these data. The following model was applied:*Y* = *μ* + *Treatment* + *Plot* {*Treatment*} + *ε*
where *Y* is the abundance of a phytophagous, *Treatment* (two levels: *T. harzianum* and control) is the main effect, and the *Plot* is nested within the *Treatment* (three levels/treatment).

Row data used in the analyses of the arthropod community on the tomato leaves were the number of insects per plant sampled over time. To test whether *T. harzianum* modulated the arthropod community on tomato leaves, a Poisson generalized linear mixed model (GLMM) with a log-link function fitted with ML (maximum likelihood) and Laplace approximation was used. The Poisson distribution best approximated the process that generated the observed data since it is a discrete distribution that measures the probability that a given number of events occur in a specified period. The *p*-values for differences between the treatments, sampling dates, and their interactions were obtained through analyses of deviance (type III Wald chi-square tests). The following general model was applied:*Y* = *μ* + *Treatment* + *Date* + *Treatment × Date* + *Plot* {*Treatment* {*Date*}} + *ε*
where *Y* is the studied group of arthropods with a Poisson distribution, *Treatment* and *Date* (six levels: 12 and 22 July, 3 and 26 August, 8 and 21 September) are the fixed factors, and *Plot* is the random effect consisting of the three experimental plots nested in *Treatment* and *Date*. This model accounts for the non-independence of the data (pseudoreplication of measures) due to the different experimental plots (the random effect) that were part of the present design.

Data of the mean number of tomato fruit (marketable, rotten, and insect-damaged fruit) were also analyzed using Poisson GLMMs with *Treatment* and *Date* as fixed effects and *Plot* as a random effect nested in *Treatment* and *Date*. The data on fruit weight, length, and width were analyzed by linear mixed-effects models (LMMs) fitted with REML (restricted maximum likelihood). The homoscedasticity and normality assumptions for ANOVAs were checked and met on these data. *p*-values for differences between treatments, sampling dates, and their interactions were obtained through ANOVAs (type III Wald F tests using the Kenward–Roger approximation for the degrees of freedom [50]). The general model applied for the analysis of the arthropod community was also applied to the analysis of tomato fruit measures, but in these analyses, the factor *Date* consisted of three levels (30 August, 8 and 21 September).

For all the analyses described so far, the model distributions were also chosen as the best fitting based on AIC criteria [51] and the full models were presented.

Additionally, a two-way permutation multivariate analysis of variance (PERMANOVA) was also presented as a supplemental analysis to test for differences between treatments, sampling dates, or their interaction. The PERMANOVA (based on 9999 permutations) was performed on arthropods grouped according to their feeding behavior using the software PAST version 4.0 [52].

Differences between treatments for the QBSar index were analyzed using a two-sample *t*-test. Differences between treatments for the presence of *Peronospora* spp. on the tomato leaves were analyzed using a Pearson’s chi-square test for independence.

All statistical analyses (except the PERMANOVA) were performed in R version 4.1.2 “Bird Hippie” [53] with the lme4 [54] and lmerTest [55] packages.

## 3. Results

In this study, only the roots of the inoculated plants showed the presence of *Trichoderma*, while in the control plants, the presence of hyphae was not observed. Furthermore, the phenotypic fruit response to inoculation is proof of fungal establishment, as well as the differences in *Peronospora* spp. and in the associated community of arthropods that were collected at the same time from each treatment.

### 3.1. First Month after Transplantation: Seedling Growth Phase

During the first month after transplanting, the presence/absence of phytophages was investigated via direct observation of the seedlings. Since the sampling method was not destructive and the plants were very small, an accurate quantitative measurement of the number of insects was not possible. At this time, we mainly found winged morphs of aphids and several specimens of a beetle identified as *Chaetocnema tibialis* Illiger (Coleoptera: Chrysomelidae). The aphid species were identified as *Macrosiphum euphorbiae, Aphis craccivora* Koch, and *Aphis gossypii* Glover (Hom., Aphididae). No aphid colonies were detected at this stage. Since *A. gossypii* was only observed on two plants during the first sampling date, it was excluded from subsequent analyses. The percentage of tomato seedlings with *M. euphorbiae, A. craccivora*, and *C. tibialis* sampled on 22 June and on 1 July (that is, during the vegetative growth stage) is reported in Figure 2.

The binomial GLMs performed on the presence/absence of *A. craccivora*, *M. euphorbiae*, and *C. tibialis* showed no significant differences between treatments (*T. harzianum* vs. control), between the two sampling periods (except for *M. euphorbiae*), nor in their interaction. The number of tomato seedlings with *M. euphorbiae* was higher on 22 June than on 1 July (χ^2^ = 7.16, df = 1, *p* < 0.001).

*Chaetocnema tibialis, M. euphorbiae*, and *A. craccivora* were also captured by the yellow sticky traps. From 10 June to 1 July, a total of 114 individuals of *C. tibialis,* 266 of *A. craccivora,* and 14 of *M. euphorbiae* were collected in yellow sticky traps. Because of the low number of *M. euphorbiae* specimens captured, this species was excluded from the statistical analysis. The mean numbers of *C. tibialis* and *A. craccivora* caught per trap/treatment are shown in Figure 3.

For both *A. craccivora* and *C. tibialis*, no significant differences were found between treatments (F_1,12_ = 0.82 and F_1,12_ = 0.9, respectively) nor between experimental plots within treatments (F_4,12_ = 1.77 and F_4,12_ = 0.87, respectively). Even if not significant, the yellow sticky traps placed in the *Trichoderma* plots captured more *A. craccivora* and *C. tibialis* than the traps placed in the control plots.

### 3.2. Second–Fourth Month after Transplantation: Vegetative Growth, Flowering, Fruit Set, and Fruit Ripening

At this stage, since the tomato plants were large enough to tolerate the shedding of a few leaves, leaf samples were collected for observation in the laboratory (see Section 2 Materials and Methods). During this sampling period, 2473 arthropod specimens were collected, of which, 1108 and 1365 were obtained from plants with and without *T. harzianum* T22, respectively. The collected arthropods were grouped according to the following categories: natural enemies, piercing-sucking insects, chewing insects, and mites. We identified four families of piercing-sucking insects (Appendix A), two families of chewing insects (Appendix A), and four families of natural enemies of herbivores (predators and parasitoids, Appendix A). Among the phytophagous mites, only the two-spotted spider mite, *Tetranychus urticae* Koch (Acari: Tetranychidae) was recorded. The PERMANOVA indicated that the abundance of these arthropod groups was affected by the sampling date (F_5,348_ = 55.2, *p* < 0.001) but not by the treatment (F_1,348_ = 2.01, *p* = 0.14). More interestingly, the interaction “*Treatment*
*× Date*” was found significant (F_5,348_ = 2.7, *p* < 0.05), indicating that the arthropod community was differently affected by inoculation with *T. harzianum* T22 in relation to the sampling period.

#### 3.2.1. Piercing-Sucking Herbivores

The piercing-sucking insects collected on tomato leaves belonged to the families Aphididae (two species identified: winged morph of *A. craccivora* and apterous morph of *M. euphorbiae*), Cicadellidae, Thripidae, and Pentatomidae (identified as eggs). The abundance of insects in the piercing-sucking group is shown in Figure 4. The GLMM shows that the abundance of the piercing-sucking community was affected by the sampling dates (χ^2^ = 48.1, df = 5, *p* < 0.001) but not by the inoculation of *T. harzianum* T22, although the probability value was close to being statistically significant (χ^2^ = 3.48, df = 1, *p* = 0.06), nor by the interaction *Treatment*
*×*
*Date* (χ^2^ = 6.69, df = 5, *p* = 0.25). It is interesting to note that during the first sampling date (12 July—flowering stage), a general increase in the abundance of piercing-sucking insects was observed on the control tomato plants. During subsequent sampling dates, the abundance of piercing-sucking arthropods was extremely low.

#### 3.2.2. Chewing Insects

The chewing insects collected on tomato leaves belonged to the families of Noctuidae (identified as eggs and larvae) and Chrysomelidae (one species: *Chaetocnema tibialis*). The abundance of insects in the chewing group over time is shown in Figure 5.

The GLMM performed on the whole chewing insect community indicated that its abundance was not affected by the inoculation of *T. harzianum* T22 (χ^2^ = 0.75, df = 1, *p* = 0.38) and by the sampling dates (χ^2^ = 9.1, df = 5, *p* = 0.1). Although the probability value was close to the threshold, the interaction *Treatment*
*×*
*Date* was not significant (χ^2^ = 9.8, df = 5, *p* = 0.08). In general, the abundance of chewing arthropods increased (even if not statistically significant) in plants inoculated with *T. harzianum* T22 compared with control ones (this trend was particularly evident on 8 September).

#### 3.2.3. Natural Enemies of Insects

The natural enemies collected on tomato leaves belonged to the families of Syrphidae (identified as eggs or larvae), Braconidae (identified as mummies), Trichogrammatidae (adults), and Miridae (adults). We also identified two individuals belonging to the order of Araneae and one mite belonging to the family of Phytoseiidae. The abundances of natural enemies of herbivores over time are shown in Figure 6.

The GLMM performed on the entire community of natural enemies indicated that its abundance was affected by the sampling dates (χ^2^ = 12.9, df = 5, *p* < 0.032), but not by *T. harzianum* (χ^2^ = 0.25, df = 1, *p* = 0.62) or by the interaction *Treatment*
*×*
*Date* (χ^2^ = 4.7, df = 5, *p* = 0.45). In general, the abundance of predators and parasitoids was higher in July and then decreased in the following months.

#### 3.2.4. Spider Mites

*Tetranychus urticae* was detected on tomato leaves only from 26 August to 21 September (Figure 7).

The GLMM performed on *T. urticae* indicated that its abundance was affected by inoculation with *T. harzianum* T22 (χ^2^ = 6.9, df = 1, *p* < 0.01) and by *Date* (χ^2^ = 39.9, df = 2, *p* < 0.001); also, the interaction *Treatment*
*×*
*Date* was significant (χ^2^ = 9.6, df = 2, *p* < 0.01). For this analysis, the *Date* factor consisted of only three levels (fruit development and fruit ripening stages: 26 and 8 August, 21 September).

#### 3.2.5. Leaf Miners

The mean number of leaf mines per plant observed in the tomato experimental field during the sampling period is reported in Figure 8. Both mines of *Tuta absoluta* (Meyrick) (Lepidoptera, Gelechiidae) and *Liriomyza trifolii* Burges (Diptera, Agromyzidae) were found.

The GLMM performed on leaf miners indicated that their abundance was affected by *T. harzianum* T22 (χ^2^ = 8.5, df = 1, *p* < 0.01). The sampling dates and the interaction *Treatment*
*×*
*Date* were not significant (χ^2^ = 5.3, df = 5, *p* = 0.38 and χ^2^ = 10.5, df = 5, *p* = 0.06, respectively). The leaf miners were more numerous on the inoculated tomato plants, especially during the first sampling date, then their number decreased.

### 3.3. QBSar

The soil samples contained several microarthropods belonging to the orders Collembola, Isopoda, Protura, and Diplura, and the class Arachnida. The sporadic presence of holometabolous insect larvae was also recorded. The differences in QBSar indexes calculated for the soil from control and *Trichoderma* T22 treated plots were not statistically significant (82 ± 20 and 62 ± 5, respectively; *t*_4_ = 1.9, *p* = 0.13).

### 3.4. Crop Sampling

#### 3.4.1. Number of Fruit per Plant

Figure 9 shows the mean number of fruit harvested per plant for the two experimental treatments during the three sampling dates. Immediately after harvesting, the tomatoes were divided into three categories: marketable, rotten, and insect-damaged fruit.

The GLMM indicated that the numbers of marketable, rotten, and insect-damaged tomato fruit were affected by the sampling dates (χ^2^ = 29.3, χ^2^ = 11.8, and χ^2^ = 17.6, respectively; df = 2, *p* < 0.01 in all cases). The interaction *Treatment*
*×*
*Date* was never found to be significant. Differences between treatments were found only for rotten tomato fruit (χ^2^ = 13.7, df = 1, *p* < 0.001). In general, the number of rotten fruit was a little higher in the *T. harzianum* T22-inoculated plants, especially during the first sampling date.

#### 3.4.2. Weight, Length, and Width of Marketable Tomato Fruit

Figure 10 shows the mean values of the weight, length, and width of tomato fruit harvested from plants inoculated with *T. harzianum* T22 and the controls on the three sampling dates. For the fruit weight, statistically significant differences were only found between treatments (F_2,12_ = 8.57, *p* < 0.05), indicating that plants inoculated with *T. harzianum* T22 produced larger fruit over time.

The length and width of the tomato fruit were statistically different between dates (F_2,12_ = 45.9 and F_2,12_ = 19.76, respectively; *p* < 0.001 in both cases) and between treatments (F_1,12_ = 9.86 and F_1,13_ = 12.26, respectively; *p* < 0.01 in both cases). No significant interaction *Treatment*
*×*
*Date* was found for length (F_2,12_ = 0.39, *p* = 0.69) and width (F_2,12_ = 2.29, *p* = 0.14) of tomato fruit. As for fruit weight, plants inoculated with *T. harzianum* T22 produced longer and wider fruit. It is interesting to note that the fruit shape of the San Marzano Kero cultivar changes with time, becoming rounder and more flattened toward the end of the production season.

#### 3.4.3. Presence/Absence of Downy Mildew

At the beginning of August, all tomato plants observed in the field were in good health, despite the presence of downy mildew. Microscope observations in the laboratory performed on symptomatic leaves confirmed the presence of *Peronospora* spp. Most of the tomato leaves observed in both the control and treated plants showed only initial symptoms of downy mildew and, consequently, were less damaged. Approximately 46% of the control plants (139 out of 302) showed the initial presence of downy mildew, while only approximately 25% (48 out of 193) of the inoculated plants showed similar symptoms; this difference was statistically significant (χ^2^ = 22.4, df = 1, *p* < 0.001).

## 4. Discussion

Knowledge of the adverse effects of agrochemicals on human health and the environment has led to the search for environmentally friendly methods to control plant diseases and pests. Beneficial soil microbes promoting plant defenses, such as non-pathogenic bacteria [56], mycorrhizal fungi [57,58,59], and plant-growth-promoting fungi [18,60], are a possible alternative to pesticides. Fungi belonging to the genus *Trichoderma* are known as plant growth promoters and control agents against plant pathogens [18]. *Trichoderma* spp. potentiate and stimulate plant defense responses against plant pathogens [61]. Furthermore, *Trichoderma* fungi can directly antagonize plant pathogens through competition, antibiosis, and mycoparasitism [18,62]. After a plant is attacked by pathogenic microorganisms or herbivorous arthropods, defense mechanisms involving signal transduction pathways responding to the phytohormones salicylic acid (SA), jasmonic acid (JA), and ethylene (ET) are activated [20,21,23,63]. Plant responses induced by herbivores depend on the mode in which these organisms attack the plant, with differences between piercing/sucking and chewing organisms [28,29]. For example, the JA signaling pathway is activated in response to chewing insects, such as caterpillars, whereas piercing-sucking insects, like aphids, induce SA-related defenses [64]. SAR signaling is mainly mediated by SA-derived compounds, while ISR is regulated by the antagonistic JA/ET signaling, but dependence on SA signaling was also reported [65,66,67]. However, although the pathways regulated by JA and ET are mutually antagonistic, their synergistic interactions play a fundamental role in the ISR activation, but how plants coordinate these complex interactions is still unclear [68,69,70,71]. Although the effects of *Trichoderma* spp. against plant pathogens are widely documented in the laboratory and the field, fewer and mostly recent studies have addressed their effects on insect pests. Laboratory studies showed that *Trichoderma* spp. negatively influence both piercing-sucking [26,33,34,35,72] and chewing insects [27,33]. In addition, *Trichoderma* spp. activate the plant’s indirect defenses by attracting the natural enemies of pest insects [37,38,73]. Field studies confirming these results are almost absent. This is an important knowledge gap because the results obtained in the laboratory [74,75] may not be evident in more complex field conditions [76]. The implications from a practical perspective are significant.

Regarding *Trichoderma* spp., the only available field study investigated the effect of *T. harzianum* root inoculation on the community of pests and beneficial arthropods associated with maize foliage [40]. This study showed that, under field conditions, the abundance of piercing-sucking herbivores decreased, while that of natural enemies increased, confirming laboratory observations. In contrast, the abundance of chewing herbivores increased, and this result seemed to be inconsistent with what was previously observed in the laboratory.

Our study was the first investigation of the effects of *T. harzianum* T22 on the above-ground arthropod community in a tomato field. In addition, we investigated whether soil inoculation with *T. harzianum* T22 may have an effect on soil micro-arthropod biodiversity, the presence and degree of attack of *Peronospora* spp., and some fruit traits. This study was based on a 1-year tomato cycle in the field and, as with all experiments carried out in the field, it had intrinsic limitations due to the effects of many environmental factors that could not be controlled. Environmental variables, such as temperature, can affect the performance of *Trichoderma* spp. Temperature influences the spore germination and the hyphal growth, and, consequently, the plant colonization of the biocontrol agent. Recently, Di Lelio et al. [47] investigated the impact of temperature on the defense response induced by insects in tomato plants inoculated with T. *harzianum* T22 and *T. atroviride* P1. Tomato plants treated with T22 exhibited enhanced resistance, mediated by SA, toward *Spodoptera littoralis* and *Macrosiphum euphorbiae* at 25 °C, while *T. atroviride P1* was shown to be more effective at 20 °C [47]. Another important environmental variable is the wind. For example, depending on the turbulence and wind speed, plant VOCs rapidly become diluted within and above the plant canopy in agroecosystems [77], limiting plant protection against insects. Other factors, such as the genotype of the plant [78], may also significantly influence the trophic interaction. However, our results provide valid information on the complex plant–arthropod–microorganism interactions that occur during a season and can be fundamental to the correct development of sustainable organic agricultural systems.

During the first month after transplanting, the tomato plants were colonized by *C. tibialis* adults and winged aphids of different species (no aphid colonies were observed at this time). The colonization of tomato plants by these species was not significantly different from the control when compared to the treatment with *Trichoderma* T22. This seemed to indicate that inoculation with *T. harzianum* did not alter plant attractiveness to aphids and flea beetles in the field. The trap catches confirmed this information. Traps placed inside the *T.*-*harzianum*-T22-inoculated plots captured more aphids and beetles than those placed in the control plots, but the differences were not statistically significant.

In the following months, the foliage arthropod communities were quantified. We found that the total numbers of arthropod pests and natural enemies sampled on treated and control plants were almost the same. Unlike laboratory studies, these neutral effects are widespread in ecological above-below-ground field experiments [79], where there are many variables (and all their interactions) that can influence the final results. However, we observed that the effects of *T. harzianum* T22 on insect abundance interacted with the sampling period. This result could be related to the different attractiveness levels of the plants with *T. harzianum* as seen by a particular arthropod group [37,40,74] and the period of its appearance. To better understand the plant-mediated influence of *T. harzianum* T22 on these dynamics, the sampled arthropods were grouped into three groups: piercing-sucking or chewing herbivores and natural enemies of pests.

Our results indicated that the arthropod composition was affected differently by inoculation with *T. harzianum* T22 in relation to the sampling period, although the average abundance was not significantly different from the controls. Differences between different phytophagous groups, found by Contreras-Cornejo et al. [40] on maize, were confirmed in this study as trends and were highly influenced by the sampling date. This result could be related to the different attractiveness of the plants with *T. harzianum* as seen by a particular arthropod group [37,74] and the period of its appearance. A general increase in piercing-sucking insects was observed in control tomato plants during the flowering stage, while the abundance of chewing arthropods increased in plants inoculated with *T. harzianum* T22, particularly in September (i.e., during the fruit maturity stage). In our experimental field, there was a slightly greater number of Noctuidae eggs in treated plants, probably due to the insect deposition preference. *Trichoderma* spp. is known to promote plant N uptake [14]. As a consequence, it is possible that chewing arthropods prefer hosts with high nutritional value for mating, oviposition, and a food source for offspring [40].

Only the abundances of the spider mite *T. urticae* and the leaf miners were significantly different on plants inoculated with *Trichoderma* T22 compared to control plants. The increased abundance of leaf mines on *Trichoderma*-inoculated plants is consistent with the increase in chewing insects observed by Cornejo et al. [40]. Equally, the reduction in *T. urticae* abundance on *Trichoderma*-inoculated plants was consistent with the reduction in piercing-sucking herbivores reported by Cornejo et al. [40], although spider mites, compared to other piercing-sucking herbivores, such as aphids and whiteflies, produce substantial cellular damage.

To our knowledge, this is one of the first studies investigating the effects of *Trichoderma* plant inoculation on *T. urticae*. Until now, it has only been reported that the secondary metabolites of *Trichoderma* caused a high mortality rate in *T. urticae* [36] and future studies should be performed to assess the effects of *Trichoderma* on the performance of *T. urticae*.

No significant effects on natural enemies of insects and the QBSar index were apparent from our study. In general, the number of these arthropods was not affected by *T. harzianum*, probably as a consequence of their relatively low number of uncontrolled environmental variables operating in the field. However, even if not significant, the number of natural enemies of insects increased on 3 August. In this period, the numbers of Syrphidae eggs and Syrphidae larvae were very high, probably due to the presence of aphids.

During the first sampling date, the number of rotten fruit was slightly higher in plants inoculated with *T. harzianum* T22, but the number of marketable fruit was almost the same between treatments. In addition, the weight of marketable fruit in the inoculated plants was increased by about 20%. Taking these results together, the production of fresh marketable tomatoes in the field could increase with *T. harzianum* T22 inoculation. Our study also confirmed the beneficial effect of *T. harzianum* T22 against plant pathogens. Downy mildew was recorded in both the control and treated tomato plants. However, even if the control tomato plants were more attacked by *Peronospora* spp. (46%) than plants treated with the *T. harzianum* T22 strain (25%), in both cases, the disease did not develop further. It was shown that *Trichoderma* positively influences tomato fruit yields as a consequence of enhanced plant growth [80]. The fungus increases the availability of nutrients to the host plant, lowers the ethylene level within the plant, and enhances the production of stimulatory compounds, such as plant growth regulators [80]. Our results confirmed that *Trichoderma* T22 inoculation in tomato has the potential to improve fruit yields.

## 5. Conclusions

This is one of the few studies that explored the potential beneficial effects related to the use of the *Trichoderma harzianum* strain T22 in tomato under field conditions. The interaction of the complex of the *T. harzianum*/tomato plant with arthropods appeared to be complex. Our results confirmed that inoculation with *T. harzianum* T22 could alter the arthropod community and reduce the abundance of specific pests under field conditions. The discrepancies between the results obtained in the laboratory and the field should be better investigated to understand their causes. For example, it would be necessary to know whether interactions with a herbivorous insect change with its developmental stage and how the foraging behavior of the adult is affected. Finally, our study suggested the possibility of using *Trichoderma* as a green alternative agent in agriculture.

## Figures and Tables

**Figure 1 insects-13-00418-f001:**
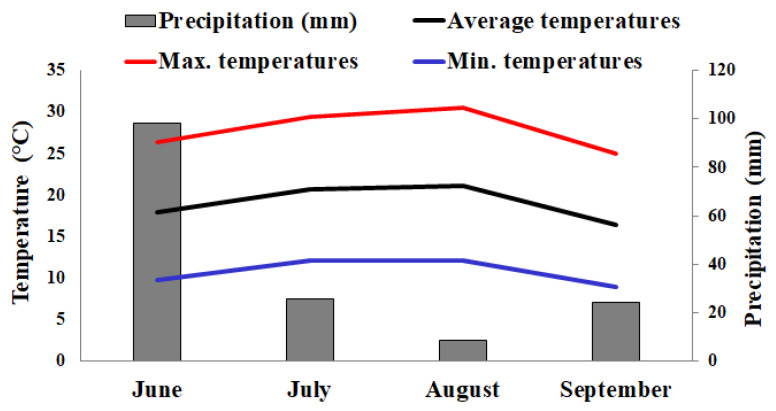
Meteorological data for Pignola, Potenza, Italy, registered from June to September 2021. Data were provided by the Agrometeorological Service of the Basilicata Region, Italy. June was the month when the *T*. *harzianum* T22 inoculation and transplantation of tomato seedlings were carried out; July: vegetative growth and flowering stages; August: stages of establishment and development of fruit; September: fruit maturity stage and harvesting of tomato fruit.

**Figure 2 insects-13-00418-f002:**
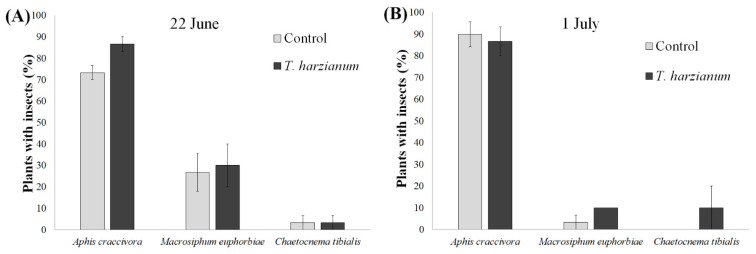
Presence of phytophages on tomato seedlings. Mean values (±standard errors) of the percentage of tomato seedlings showing the presence of phytophagous insects, as recorded on 22 June (**A**) and on 1 July (**B**).

**Figure 3 insects-13-00418-f003:**
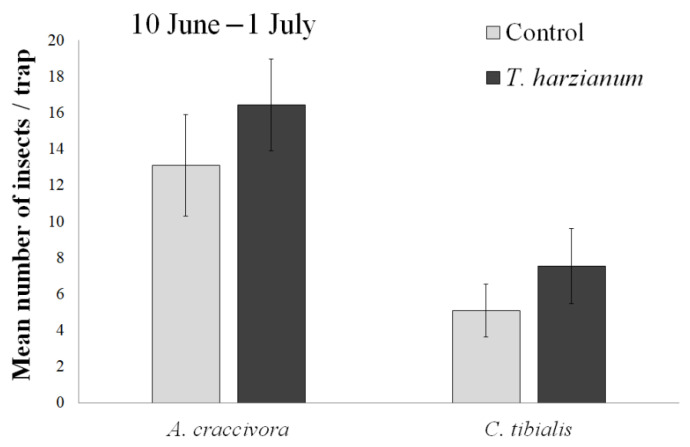
Presence of phytophages on yellow sticky traps. Mean values (± standard errors) of *A. craccivora* and *C. tibialis* individuals collected per trap for each treatment from 10 June to 1 July.

**Figure 4 insects-13-00418-f004:**
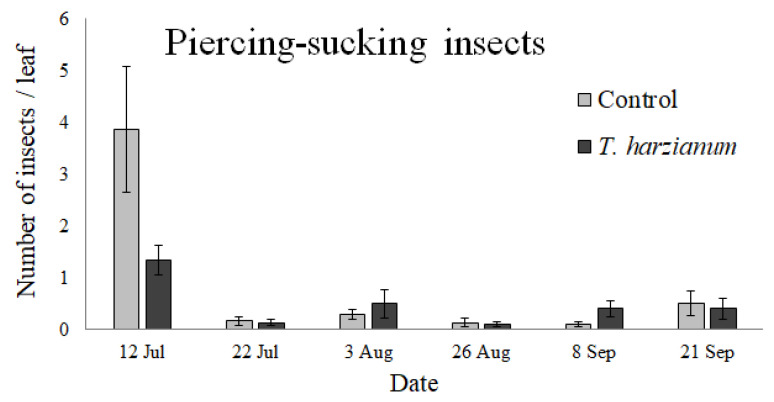
Mean values (±standard errors) of piercing-sucking insects collected on tomato leaves during the experiment.

**Figure 5 insects-13-00418-f005:**
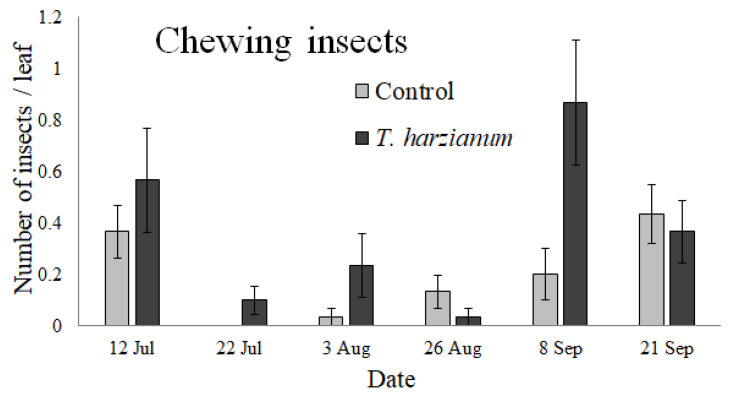
Mean values (±standard errors) of insect abundance in the chewing group collected on tomato leaves during the experiment.

**Figure 6 insects-13-00418-f006:**
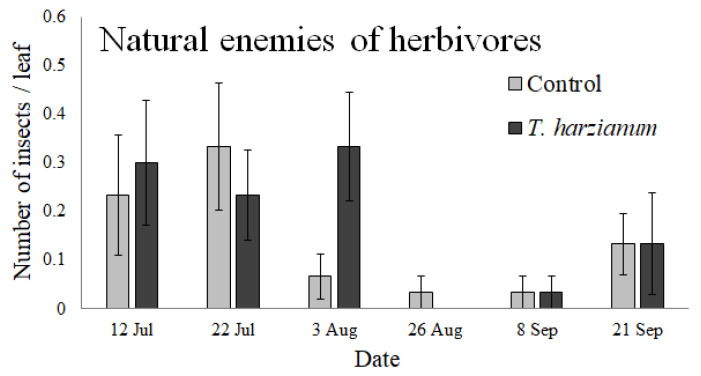
Mean values (±standard errors) of natural enemies of herbivores (predators and parasitoids) collected on tomato leaves during the experiment.

**Figure 7 insects-13-00418-f007:**
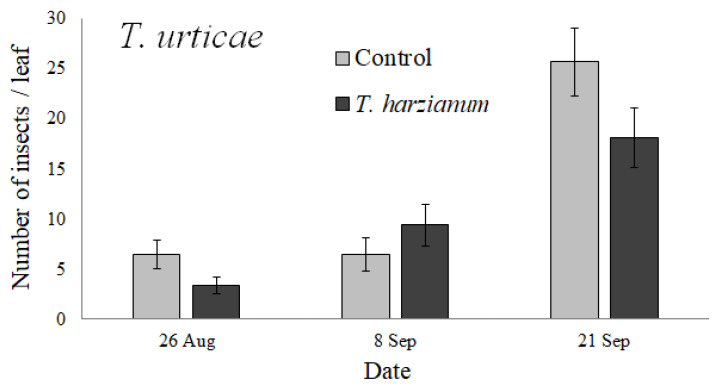
Mean values (±standard errors) of the abundance of *Tetranychus urticae* collected on tomato leaves during the experiment.

**Figure 8 insects-13-00418-f008:**
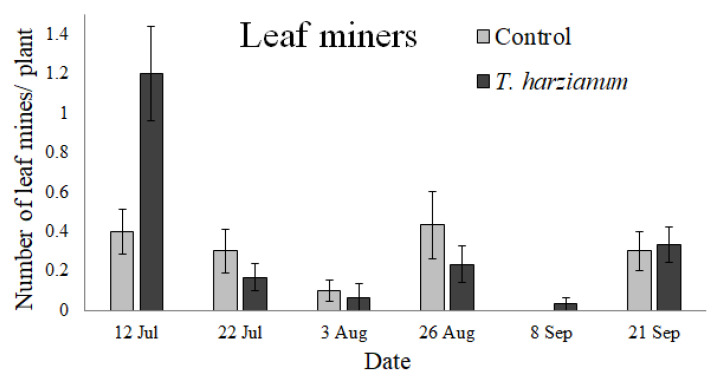
Mean values (±standard errors) of insects that cause leaf mine damage per plant associated with the tomato leaves over time.

**Figure 9 insects-13-00418-f009:**
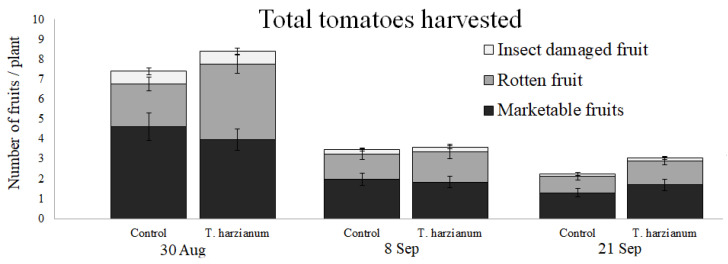
Mean number of tomato fruit (±standard errors) harvested per plant in the two experimental treatments during the three sampling dates.

**Figure 10 insects-13-00418-f010:**
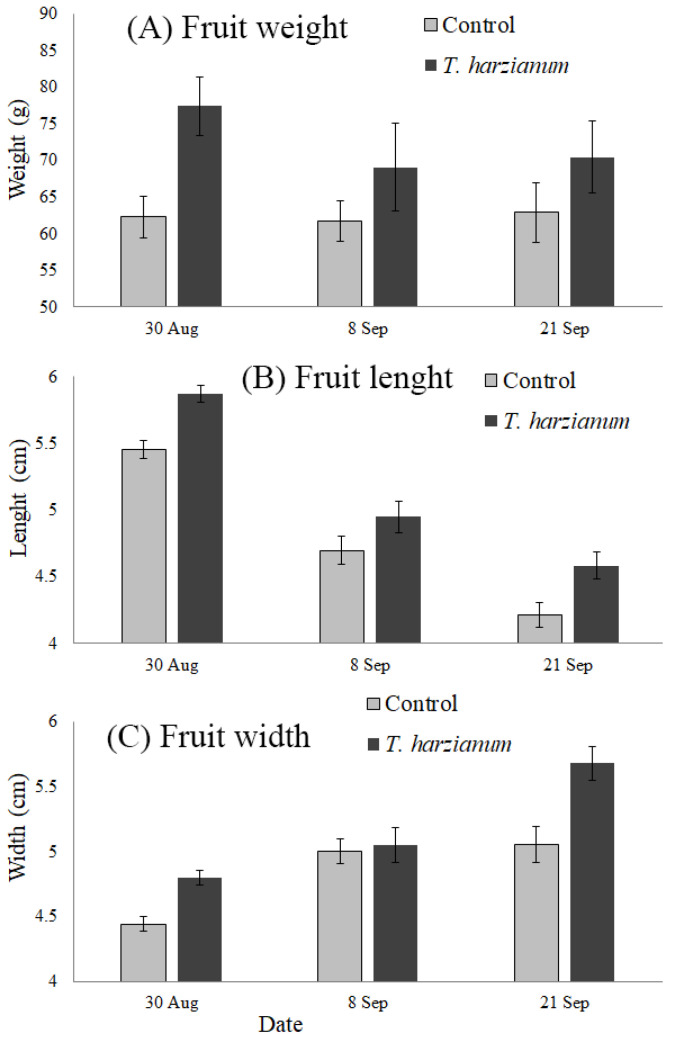
Mean values (±standard errors) of fruit weight, length, and width from tomato plants inoculated with *T. harzianum* T22 and control ones at the three sampling dates.

## Data Availability

The data presented in this study are available from the corresponding author upon request.

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
