# Peer review of "Effects of Trichoderma harzianum Strain T22 on the Arthropod Community Associated with Tomato Plants and on the Crop Performance in an Experimental Field"

_insects, 2022, doi:10.3390/insects13050418_

Round 1

Reviewer 1 Report

Line 18: The use of Trichoderma in open field conditions has been wide reported, in fact, there are commercial products in the market worldwide. This is also contrary to the claim in lines 24-25.

Lines 70-89: This information is valuable if the action of Trichoderma will be associated to these responses. Please consider to reduce this text if there is not a clear relation in this work. Alternatively, this could be dealt with at the Discussion section.

Lines 178-185: Give a reference of previous work where this fertigation scheme was taken from. Was this developed during the present work?

Lines 371-372: M. euphorbiae presented significant differences between treatments, What about Chaetocnema tibialis on July 1st?

Lines 434-440: The sample analysed on the 8th September showed a higher number of chewing insects in the treatment with Trichoderma. How can this be explained? Please comment since there is not any relation to the presence of natural enemies of herbivores (Fig. 6)

Line 454: Although this claim does not apply to samples from August 3rd. Explain.

Line 505: This would not represent an advantage for the use of Trichoderma. Would Trichoderma be recommendable anyway? Please comment.

Lines 549-550: This is a rather poor contribution since the performance of Trichoderma in open fields has been reported, as a realistic alternative already. In fact, this sustains commercial formulations with this fungus.

Lines 573-575: Indeed, this is a valuable information, nonetheless authors should explain rather than merely list the results obtained, as described in lines 591-593 based on bibliographic analysis. Any correlation within variables?

Lines 599-601: Authors compare results to similar publications, which is fine. Some clear contributions should focus on the explanations of these results correlating with other variables. Again, confirmation of previous results renders a limited novelty of this manuscript and it is recommended to find new interactions among variables of new interpretations based on bibliographic analysis

Lines 620-629: This paragraph is a nice example of how discussion of results should be accomplished, based on bibliographic analysis and avoiding a repetition of results in this section.

Lines 639-641: These claims are not Conclusions of the work, they could be part of the Discussion

Author Response

Reviewer 1

We thank the Reviewer 1 for her/his very useful comments and efforts towards improving our manuscript. The following changes have been done as requested by the reviewers:

 Line 18: The use of Trichoderma in open field conditions has been wide reported, in fact, there are commercial products in the market worldwide. This is also contrary to the claim in lines 24-25.

RESPONSE. As suggested by the reviewer, we delete this sentence.

Lines 70-89: This information is valuable if the action of Trichoderma will be associated to these responses. Please consider to reduce this text if there is not a clear relation in this work. Alternatively, this could be dealt with at the Discussion section.

RESPONSE. As suggested by the reviewer, part of this paragraph has been moved to the Discussion.

Lines 178-185: Give a reference of previous work where this fertigation scheme was taken from. Was this developed during the present work?

RESPONSE. As suggested by the reviewer, this sentence has been added in Materials and Methods: “The use of fertilizers was developed during the present experiment and was appropriate to the needs of the plants, considering the soil characteristics.”

Lines 371-372: M. euphorbiae presented significant differences between treatments, What about Chaetocnema tibialis on July 1st?

RESPONSE. As stated in the text, the presence/absence of C. tibialis showed no significant differences between treatments on 1 July. Individuals of M. euphorbiae were present in all plots inoculated with T. harzianum, while individuals of C. tibialis were observed only in one plot. The sporadic presence of C. tibialis among plots therefore makes the differences between treatments not significant. This behavior can be seen by observing the standard errors in figure 2 during 1 July (zero for M. euphorbiae, 10 for C. tibialis).

Lines 434-440: The sample analysed on the 8th September showed a higher number of chewing insects in the treatment with Trichoderma. How can this be explained? Please comment since there is not any relation to the presence of natural enemies of herbivores (Fig. 6)

RESPONSE. As stated in the text, the interaction “treatment x date” was not significant, although the probability value was close to the threshold. As suggested by the reviewer we tried to explain this trend in the Discussion section:

“In our experimental field, there was a slightly greater number of Noctuidae eggs in treated plants, probably due to the insect deposition preference. Trichoderma spp. is known to promote plant N uptake (Harman et al., 2004). As a consequence, it is possible that chewing arthropods prefer hosts with high nutritional value for mating, oviposition, and food source for offspring (Contreras-Cornejo et al., 2020)”.

Line 454: Although this claim does not apply to samples from August 3rd. Explain.

RESPONSE. As stated in the text,  the abundance of natural enemies was affected by the sampling dates but not by T. harzianum or by the interaction “treatment X date”. As suggested by the reviewer we tried to explain this trend in the Discussion section: “…. However, even if not significant, the number of natural enemies of insects increased on 3 August. In this period, the number of Syrphidae eggs and Syrphidae larvae were very abundant, probably due to the presence of aphids.”

Line 505: This would not represent an advantage for the use of Trichoderma. Would Trichoderma be recommendable anyway? Please comment.

RESPONSE. As suggested by the reviewer, we tried to explain these results in the Discussion section:

“During the first sampling date, the number of rotten fruits was slightly higher in plants inoculated with T. harzianum T22, but the number of marketable fruits was almost the same between treatments. Furthermore, the weight of marketable fruit in the inoculated plants is increased by about 20%. Taking these results together, the production of fresh marketable tomatoes in the field can increase with inoculation with T. harzianum T22.”

Lines 549-550: This is a rather poor contribution since the performance of Trichoderma in open fields has been reported, as a realistic alternative already. In fact, this sustains commercial formulations with this fungus.

RESPONSE. As suggested by the reviewer, we delete this sentence.

Lines 573-575: Indeed, this is a valuable information, nonetheless authors should explain rather than merely list the results obtained, as described in lines 591-593 based on bibliographic analysis. Any correlation within variables?

RESPONSE. As suggested by the reviewer, we add this information:

“Environmental variables such as temperature can affect the performance of Trichoderma spp. Temperature influences the spore germination and the hyphal growth, and, consequently, the plant colonization of the biocontrol agent. Recently, Di Lelio et al. (2021) investigated the impact of temperatures on the defense response induced by insects in tomato plants inoculated with T. harzianum T22 and T. atroviride P1. Tomato plants treated with T22 exhibited enhanced resistance, mediated by SA, toward Spodoptera littoralis and Macrosiphum euphorbiae at 25°C, while T. atroviride P1 was shown to be more effective at 20°C (Di Lelio et al., 2021). Another important environmental variable is the wind. For example, depending on the turbulence and wind speed, plant VOCs rapidly become diluted within and above the plant canopy in agroecosystems (Loivamäki et al., 2008), limiting plant protection against insects.”

Lines 599-601: Authors compare results to similar publications, which is fine. Some clear contributions should focus on the explanations of these results correlating with other variables. Again, confirmation of previous results renders a limited novelty of this manuscript and it is recommended to find new interactions among variables of new interpretations based on bibliographic analysis

RESPONSE. As suggested by the reviewer, we tried to explain why we find differences in the arthropod community based on the sampling period. We add this sentence:

“This result could be related to different attractiveness of the plants with T. harzianum towards a particular arthropod group (Guerrieri et al., 2004; Battaglia et al., 2013) and to the period of its appearance.”

See also the responses to previous comments.

Lines 620-629: This paragraph is a nice example of how discussion of results should be accomplished, based on bibliographic analysis and avoiding a repetition of results in this section.

RESPONSE. Thank you.

Lines 639-641: These claims are not Conclusions of the work, they could be part of the Discussion.

RESPONSE. As suggested by the reviewer, this sentence has been moved to the Discussion section.

Reviewer 2 Report

Comment on Article submitted to Journal Insects

Effects of Trichoderma harzianum Strain T22 on the Arthropod Community Associated with Tomato Plants and on the Crop             Performance in an Experimental Field

Vittoria Caccavo 1 , Pierluigi Forlano 1 , Stefania Mirela Mang 1 , Paolo Fanti 2 , Maria Nuzzaci 1 , Donatella Battaglia 1 5 and Vincenzo Trotta 1,*

Aim of the submitted article is to investigate the effects of the inoculation of a commercial Trichoderma harzianum strain T22 on the arthropod community associated with tomato plants in an experimental field located in South Italy. Looked was at the beneficial effects of T. harzianum on soil arthropod biodiversity, the agronomic performance of tomato plants (yield) and on the behaviour toward downy mildew (tomato phytopathogen). Discussed is the in the field occurring complex tomato–arthropod–microorganism interactions and the use of Trichoderma as a green alternative in agriculture.

From soil samples over the tomato growing period extracted were microarthropods and evaluated downy mildew on tomato plants, the agronomic performance estimated and statistical analysed by the model Y = μ + Treatment + Date + Treatment × Date + ε , a linear, maximum likelihood fitted mixed model (GLMM ANOVA, ), and Peronospora spp. on the tomato leaves by Pearson's chi-square test for Independence.

That the fungal genus Trichoderma has a plant growth promoting and controlling potential against plant pathogens has been observed before and consequently the paper as part of a research project, titled "Sustainable control of insect vectors of phytopathogenic viruses in the context of climate change: the role of root symbionts" has tested efficiency related Trichoderma under field conditions in a tomato plantation. In spite of the finding that by sampling the total number of arthropod pests and natural enemies on treated and control plants the stocking was almost the same, an inoculation with T. harzianum T22 can alter the arthropod community in consistency with the increase in chewing, the abundance of specific pests under field conditions reducing insects. Significantly different on plants inoculated with Trichoderma T22 compared to control plants were the abundances of the spider mite T. urticae and the leaf miners (natural enemies on tomato leaves: Syrphidae (identified as eggs or larvae), Braconidae (identified as mummies), Trichogram matidae (adults), and Miridae (adults), and of 2 individuals belonging to the order of Araneae and 1 mite belonging to the family of Phytoseiidae). In the soil samples were several to the orders of Collembola, Isopoda, Protura, Diplura, and class of Arachnida belonging microarthropods (no difference after treatment) and the chewing insects collected on tomato leaves belonged to the families of Noctuidae (identified as eggs and larvae) and Chrysomelidae (one species: Chaetocnema tibialis) were not affected by the inoculation of T. harzianum. Downy mildew concerned plants showed only initial symptoms and plants inoculated with T. harzianum T22 produce larger fruits over time, though the number of rotten fruits was a little higher in the of T. harzianum T22 inoculated plants.

Since field studies confirming these results are almost absent, as is said, I recommend to publish the little corrections needing paper, the more because the whole concept focusses on phytopathogenic viruses, carried by all organisms involved, including T. harzianum T22. Lysogenic, lytic replicating viruses are important collectors of epigenetic information, which is transferred in between the parties hereto.

Author Response

Reviewer 2

We would like to thank the Reviewer 2 for her/his comments on our manuscript.

The virus transmission is an important topic in many contests related to agriculture. As reported by reviewer 2, one of us (VC) is supported by a grant on a Ph.D. project entitled “Sustainable control of insect vectors of phytopathogenic viruses in the context of climate change: the role of root symbionts”. In the present experiment, however, the presence of phytopathogenic viruses was not observed on tomato plants.